# Causal Structure Discovery between Clusters
# of Nodes Induced by Latent Factors

**Chandler Squires\***                                                    CSQUIRES@MIT.EDU
*LIDS, IDSS, and CSAIL, MIT, Cambridge, MA, USA*

**Annie Yun\***                                                          ANNIEYUN@MIT.EDU
*LIDS and IDSS, MIT, Cambridge, MA, USA*

**Eshaan Nichani**                                                       ESHNICH@MIT.EDU
*LIDS and IDSS, MIT, Cambridge, MA, USA*

**Raj Agrawal**                                                   R.AGRAWAL@CSAIL.MIT.EDU
*LIDS, IDSS, and CSAIL, MIT, Cambridge, MA, USA*

**Caroline Uhler**                                                        CUHLER@MIT.EDU
*LIDS and IDSS, MIT, and Broad Institute, Cambridge, MA, USA*

**Editors:** Bernhard Schölkopf, Caroline Uhler and Kun Zhang

## Abstract

We consider the problem of learning the structure of a causal directed acyclic graph (DAG) model in the presence of latent variables. We define *latent factor causal models* (LFCMs) as a restriction on causal DAG models with latent variables, which are composed of clusters of observed variables that share the same latent parent and connections between these clusters given by edges pointing from the observed variables to latent variables. LFCMs are motivated by gene regulatory networks, where regulatory edges, corresponding to transcription factors, connect spatially clustered genes. We show identifiability results on this model and design a consistent three-stage algorithm that discovers clusters of observed nodes, a partial ordering over clusters, and finally, the entire structure over both observed and latent nodes. We evaluate our method in a synthetic setting, demonstrating its ability to almost perfectly recover the ground truth clustering even at relatively low sample sizes, as well as the ability to recover a significant number of the edges from observed variables to latent factors. Finally, we apply our method in a semi-synthetic setting to protein mass spectrometry data with a known ground truth network, and achieve almost perfect recovery of the ground truth variable clusters.

**Keywords:** Causal discovery, causal structure learning, causal identifiability, latent factor model

## 1. Introduction

Structural causal models are valuable tools for reasoning about decision-making, and as a result, have been widely adopted across fields such as genomics (Friedman et al., 2000), econometrics (Blalock, 2017), and epidemiology (Robins et al., 2000). To use causal models when the causal structure is not known *a priori*, it is necessary to learn the model from observed data, a task known as *causal structure learning* (Heinze-Deml et al., 2018). As a field, causal structure learning has recently experienced major developments and remains an active and widespread area of research. Recent works aim to address a number of challenges inherent to the problem of learning causal

---

*Equal contribution

structure, such as the presence of unobserved confounders (Cai et al., 2019; Frot et al., 2019; Bernstein et al., 2020), the large search space over causal models (Chickering, 2002; Solus et al., 2021), identifiability of the underlying causal model (Shimizu et al., 2006; Peters and Bühlmann, 2014), and statistical issues stemming from high-dimensional datasets (Nandy et al., 2018). We focus on a setting which exhibits all of these challenges, and our proposed method addresses each of these challenges in a cohesive way. We devote particular attention to the issue of unobserved confounders,

A number of methods have been proposed to address the challenge of learning causal models in the presence of unobserved confounders. These methods fall into two general categories. First, some methods account for unobserved confounders by learning a graphical model over only the observed variables, albeit from a *different* class of graphical models (Richardson and Spirtes, 2002; Bernstein et al., 2020). However, in some cases, such as the one explored in this paper, it is possible to learn a graph over *both* the observed and latent variables. Existing methods (Silva et al., 2006; Kummerfeld and Ramsey, 2016; Xie et al., 2020; Agrawal et al., 2021) that seek to recover these structures often assume that the latent variables are *exogenous*, i.e., are not caused by any of the observed variables. However, this assumption is often violated in many applications. For example, in genomics, gene regulatory networks are often modeled using *transcription factories* (Stadhouders et al., 2019) as underlying latent variables with gene expression as the observable variables. These gene expressions then can have downstream impacts on other transcription factories, requiring a model that allows non-exogenous latent variables.

**Contributions.** In Section 2, we introduce the class of *latent factor causal models* (LFCMs), which allow for non-exogenous latent variables. Similar to prior work, this class of models prohibits direct edges between observed variables, i.e., the effect of one observed variable on another must be mediated by some latent variable. We likewise prohibit direct edges between latent variables, inducing a *bipartite* structure over the graph. Furthermore, we require that the latent variables *cluster* the observed variables, i.e., each observed variable has only a single latent parent, and that each latent variable has at least three observed children. These constraints on the model are motivated by how the DNA is organized in the cell nucleus to facilitate cell-type specific gene expression. The spatial clustering of genes in the cell nucleus facilitates their co-regulation by transcription factors (Belyaeva et al., 2017; Uhler and Shivashankar, 2017). The expression of each gene represents the observed variables, the spatial clustering of genes is unobserved, and the latent factors represent the presence of transcription factors that can e.g. turn on the expression of the co-clustered genes.

In Section 3, we establish identifiability results for LFCMs, based primarily on the tetrad representation theorem of Spirtes (2013). Based on our identifiability results, in Section 4 we propose a constraint-based method for learning the underlying graph over *both* latent and observed variables. The proposed method has three stages. In the first stage, our method identifies clusters of observed variables with the same latent parent, as well as an ordering over these clusters. The second stage merges clusters from the first stage if necessary. In the third stage, we learn edges from the observed variables to the latent variables, by testing for conditional independence with all children of each latent variable. For each stage, the constraints being checked are equivalent to multiple test statistics vanishing simultaneously, requiring the use of multiple hypothesis testing procedures which we describe in Section 4.1. Finally, in Section 5, we demonstrate the performance of our algorithm in both a completely synthetic and a semi-synthetic setting. In particular, we show that our method is capable of recovering the ground truth clustering with nearly 100% accuracy even at relatively low sample sizes. Our method also recovers the ground truth edges between observed nodes and latent nodes with higher accuracy than a baseline which does not make use of multiple hypothesis testing.

## 1.1. Related Work

**Learning undirected graphical models with clusters.** Since clusters of correlated variables are common across many disciplines, including biology (Eisen et al., 1998), economics (Bai and Wang, 2016), neuroscience (Arslan et al., 2018; Pircalabelu and Claeskens, 2020), and the behavioral sciences (van der Linden and Hambleton, 2013), several structure learning methods have been developed which encourage clustering in the estimated graphs, especially in the setting of *undirected* graphical models. For example, Tan et al. (2015) introduced the *cluster graphical lasso* method, which generalizes the traditional graphical lasso method to allow for the incorporation of known clustering information, resulting in denser estimated subgraphs over these clusters. Building on this work, Hosseini and Lee (2016) introduce the *GRAB* algorithm, which does *not* require clusters to be known beforehand, but instead allows the clustering to be learned simultaneously to network structure. More recently, Pircalabelu and Claeskens (2020) introduced *ComGGL*, a method which also learns clusters and graph structure simultaneously, with the additional benefit of high-dimensional consistency guarantees for both cluster recovery and graph structure in sparse settings.

**Latent tree models and factor analysis.** Unlike in the undirected settings above, in our setting, the clusters of observed variables are explicitly assumed to be induced by latent variables. As has been observed in previous works, especially in latent tree modeling (Choi et al., 2011; Shiers et al., 2016; Drton et al., 2017; Leung and Drton, 2018) and factor analysis (Drton et al., 2007; Kummerfeld and Ramsey, 2016), these latent variables produce "signatures" or "invariants" in the distribution over the observed variables, which can be exploited for structure learning. One invariant which plays an important role in both settings is the *tetrad* $t_{ij,uv}$, a $2 \times 2$ subdeterminant of the correlation matrix which must *vanish* (i.e., equal zero) whenever $i$ and $j$ share a single common latent parent, but have no children. As we will see in Section 3, despite the differences in our model assumptions, tetrads also play an important role in our algorithm when identifying causal clusters.

**Traditional causal discovery methods.** Within the space of discovering causal models on observational data, there are two categorizations of algorithms. First, there are constraint-based methods, which rely on conditional independence testing to draw conclusions about the structure. The well-known PC-algorithm assumes *causal sufficiency*, which bars unmeasured common cause latent variables and selection variables (Spirtes et al., 2000). There also exist constraint-based methods on directed acyclic graphs with latent and selection variables, such as FCI, RFCI and their variants (Spirtes, 2001; Colombo et al., 2012). These methods all learn Markov equivalence classes of directed acyclic graphs, as represented by completed partially directed acyclic graphs (CPDAGs) in the PC-algorithm or partial ancestral graphs (PAGs) in the FCI algorithm. The second categorization of methods is score-based algorithms, such as GES (Chickering, 2002), which identify underlying structure by optimizing a well-designed score function. These methods, even those that are asymptotically correct in the presence of latent confounders, output equivalence classes of DAGs. In our work, we try to recover more complete causal information.

**Learning causal models with latent variables.** Existing work for structure recovery in the presence of latent variables can often by characterized by the model structures that the method performs well or poorly upon. Agrawal et al. (2021) considers the model in which latent variables are *pervasive*, influencing many observed nodes. Their method, *DeCAMFounder*, recovers the true causal structure over observed variables by applying spectral decomposition in the non-linear additive noise and pervasive confounding setting (Agrawal et al., 2021), extending the linear setting of Frot et al. (2019). A number of methods, similarly to the current work, also rely on algebraic

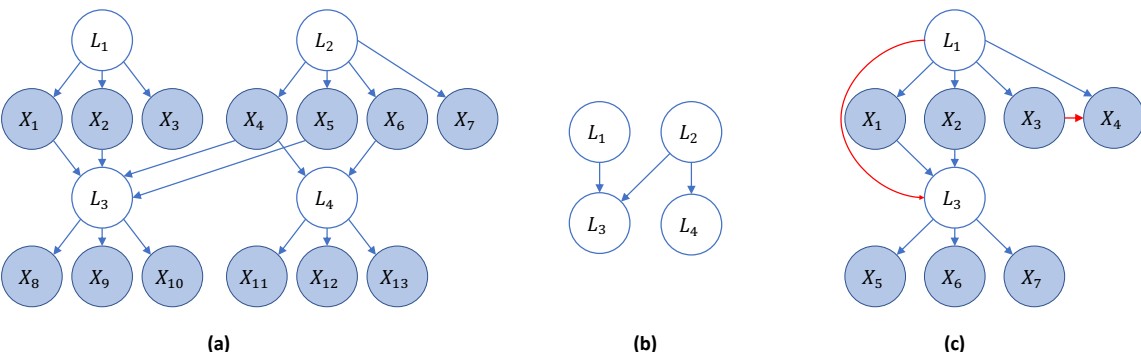

Figure 1: **(a)** $\mathcal{G}$ satisfies our model constraints. **(b)** The latent graph $L(\mathcal{G})$ for $\mathcal{G}$. **(c)** $\mathcal{G}'$ falls outside of the class of models we consider in this paper, with violations shown in red.

constraints in the covariance matrix over observable variables to infer graph structure over latent variables. The BPC (Silva et al., 2006) and FOFC (Kummerfeld and Ramsey, 2016) algorithms both leverage rank constraints on the covariance matrix to cluster observed variables, then recover some structure over the inferred latent nodes corresponding to these clusters. Other algorithms, such as those proposed by Shimizu et al. (2009), Cai et al. (2019), and Xie et al. (2020), attempt to improve upon previous algorithms by restricting models to the *linear non-Gaussian* case. In our work, we do not require non-Gaussianity, instead working in the general linear acyclic model regime. Furthermore, all of the above algorithms rely on the *measurement assumption*, which requires that no observed variable is the parent of any latent variable. This assumption, however, is not satisfied in many real-world applications of graphical models with latent variables, and in our work, we attempt to recover causal structures *without* the measurement assumption.

## 2. Problem Setup

We now formally define the class of models considered in this paper. A *structural causal model* (SCM) over the variables $\{X_i\}_{i=1}^p$ consists of a set of *structural assignments* of the form $X_i = f_i(X_{\text{pa}(i)}, \epsilon_i)$, and a product distribution $\mathbb{P}_\epsilon$ over mean-zero *exogenous noise* terms $\{\epsilon_i\}_{i=1}^p$. The set $\text{pa}(X_i)$ are called the *parents* of $X_i$, and the *causal graph* for the SCM is a graph with nodes $\{X_i\}_{i=1}^p$ and directed edges $X_j \to X_i$ for $X_j \in \text{pa}(X_i)$. We assume that the causal graph for the SCM is *acyclic*, in which case the distribution $\mathbb{P}_\epsilon$ induces a unique distribution $\mathbb{P}_X$ over $\{X_i\}_{i=1}^p$.

In this paper, we focus on a class of SCMs with restrictions on both the structural assignments and on the causal graph. First, we assume each $f_i$ is a *linear* function, a common starting point for new methods, which has been the setting of many works (Chickering, 2002; Hauser and Bühlmann, 2012; Solus et al., 2021). Second, we assume that the causal graph is of the following form.

**Definition 1** *Let $\mathcal{G}$ be a DAG over latent nodes $L_1, \ldots, L_K$ and observed nodes $X = \{X_1, \ldots, X_p\}$. The clusters of $\mathcal{G}$ are the sets $C_k = \text{ch}(L_k)$ for $k = 1, \ldots, K$. The latent graph for $\mathcal{G}$, denoted $L(\mathcal{G})$, is the graph over $\{L_k\}_{k=1}^K$ with an edge $k \to k'$ if and only if $X_k \to L_{k'}$ for some $X_k \in \text{ch}(L_k)$. $\mathcal{G}$ is called a latent factor causal model (LFCM) if it satisfies the following conditions:*

*(a) [Unique cluster assumption] Each observed node has exactly one latent parent.*

*(b) [Bipartite assumption] There are no edges between pairs of observed nodes or between pairs of latent nodes.*

*(c) [Triple-child assumption] Each latent node has at least 3 observed children.*

*(c) [Double-parent assumption] If $k \to k'$ in $L(\mathcal{G})$, then there exist two nodes $X_i, X_j \in \text{ch}(L_k)$ such that $X_i \to L_{k'}$ and $X_j \to L_{k'}$.*

See Fig. 1a for an example of a graph that satisfies our model definition, and Fig. 1c for a graph that does not. The importance of each assumption for the purpose of identifying $\mathcal{G}$ will become clear in the proofs of the genericity and identifiability results presented in the next section. For example, we will see that the edge $X_3 \to X_4$ in Fig. 1c prevents the submatrix $\Sigma_{[1,2],[3,4]}$ of the covariance matrix $\Sigma$ from being low rank, and thus prevents $\{X_1, X_2\}$ and $\{X_3, X_4\}$ from being clustered.

## 3. Trek separation and genericity assumptions

In this section, we review fundamental results essential to the identifiability of LFCMs, which we constructively prove in Section 4 by introducing an algorithm for consistently estimating LFCMs. We will also introduce genericity assumptions necessary for the consistency of our algorithm.

We denote the covariance matrix of our model as $\Sigma$, and given two subsets of nodes $A, B$, we use $\Sigma_{A,B}$ to denote the submatrix of $\Sigma$ with rows in $A$ and columns in $B$. Our identifiability results rely on a common generalization of d-separation, known as *trek separation*, which relates the causal graph of a SCM to the rank of submatrices of $\Sigma$. A *directed path* from node $i$ to node $j$ is a sequence of nodes $p_1 = i, ..., p_k = j$, such that $p_i \to p_{i+1}$ for all $i$ from 1 to $k-1$. In this case, $j$ is called the *sink* of the path and $i$ is called the *source*. A *trek* in the graph $\mathcal{G}$ from $i$ to $j$ is an ordered pair of directed paths $(P_1, P_2)$, such that the sink of $P_1$ is $i$ and the sink of $P_2$ is $j$, and $P_1, P_2$ share a source $k$. Now, we define *trek separation*. Given four subsets $A, B, C_A, C_B$ of nodes (note these subsets need not be disjoint), the pair $(C_A, C_B)$ *t-separates* $A$ and $B$ if, for every trek $(P_1, P_2)$ between $A$ and $B$, $P_1$ contains a node in $C_A$ or $P_2$ contains a node from $C_B$. Finally, the following theorem relates the notion of *t-separation* to the rank of submatrices of the covariance matrix.

**Theorem 1 (Trek separation, Sullivant et al. (2010))** *Let $A, B$ be two subset of nodes in $\mathcal{G}$. Then*

$$rank(\Sigma_{A,B}) \leq \min\{|C_A| + |C_B| : (C_A, C_B) \text{ t-separates } A \text{ from } B \text{ in } \mathcal{G}\}$$

*Moreover, equality holds generically[1] for $\Sigma$ consistent with $\mathcal{G}$.*

In this paper, we only need to use information about the rank of 2 x 2 submatrices of $\Sigma$. The determinants of these matrices are commonly known as *tetrads*. In particular, we denote $t_{ij,uv} = \det(\Sigma_{[ij],[uv]}) = \Sigma_{iu}\Sigma_{jv} - \Sigma_{iv}\Sigma_{ju}$. Specializing Theorem 1, we obtain the following corollary:

**Corollary 1 (Tetrad representation, Spirtes (2013))** *Suppose $A = \{X_i, X_j\}$ and $B = \{X_u, X_v\}$ are t-separated by a single node. Then $t_{ij,uv} = 0$.*

We can now see the importance of the first three assumptions in Definition 1. These structural assumptions control the size of t-separating sets between nodes in the same cluster and in different clusters, so that we can apply Theorem 2 and Corollary 1 to ensure that certain tetrads are

---

[1] We say a statement holds *generically* if the set of parameters for which it does not hold has Lebesgue measure zero.

either zero or generically non-zero. In particular, the unique cluster assumption and the bipartite assumption guarantees that two nodes $X_i$ and $X_j$ in the same cluster will be t-separated from their non-descendants by their latent parent. Thus, clusters of nodes with no descendants can be identified. Conversely, the triple-child assumption ensures that two nodes $X_i$ and $X_j$ that are not in the same cluster do not get clustered, since we can find a 2x2 submatrix with $i$ indexing one of the rows and $j$ indexing one of the columns that is generically of rank 2. In Appendix A, we formally state and prove that the following faithfulness assumptions are indeed generic under the first 3 structural assumptions from Definition 1:

**Assumption 1 (Cluster tetrad faithfulness)** *Suppose $X_i$ and $X_j$ are not in the same cluster. Then there exists some $\{u, v\}$ such that $t_{ij,uv} \neq 0$.*

**Assumption 2 (Parent tetrad faithfulness)** *Suppose $X_i$ and $X_j$ are in the same cluster, but $X_i$ has at least one child. Then there exists some $\{u, v\}$ such that $t_{ij,uv} \neq 0$.*

**Assumption 3 (Latent adjacency faithfulness)** *Suppose $X_i \to L_k$. Let $S_i = \mathrm{ch}(\mathrm{pa}(X_i)) \setminus \{i\}$ and $S' = \cup_{j \leq i} \mathrm{ch}(L_i)$. Then $\rho_{i,k|S_i,S} \neq 0$ for some $X_k \in \mathrm{ch}(L_k)$*

**Remark 1** *Since causal structure learning algorithms are always run in a noisy setting,* near *violations of genericity assumptions can degrade the performance of a method, as discussed by Uhler et al. (2013). In particular, the set of parameters which violate a "strong" faithfulness condition is generally a positive measure set, extending from the measure zero set where faithfulness is violated. Fortunately for the current setting, our assumptions require the existence of only a* single *entry of the underlying statistic being far from zero. Thus, the set of parameters violating the "strong" version of our faithfulness assumption is an* intersection *of the sets of parameters for which each $t_{ij,uv}$ is near zero, resulting in a smaller set. In the present work, we will not attempt to quantify the size of this set and the resulting statistical benefits, but note these as interesting directions for future work.*

## 4. Methods

Our algorithm, presented in Algorithm 1, consists of three stages. As is common in causal structure learning, we present our algorithm with implementation details of hypothesis testing abstracted away. In particular, we will assume access to two subroutines, whose implementation details will be given in Section 4.1. The first subroutine tests $H_{ci}(X_j, X_A \mid X_B)$, which

---

**Algorithm 1** `EstimateLFCM`

**Input:** Data $\mathbb{X} \in \mathbb{R}^{n \times p}$.
Let $\pi = $ `FindOrderedClusters`$(\mathbb{X})$
Let $\pi = $ `MergeClusters`$(\mathbb{X}, \pi)$
Let $\hat{\mathcal{G}} = $ `LearnDAG`$(\mathbb{X}, \pi)$
**Output:** $\hat{\mathcal{G}}$

---

denotes the null hypothesis that $X_j$ and $X_A$ are conditionally independent given $X_B$. The second subroutine tests $H_{vt}(X_A, X_B)$, which denotes the null hypothesis that all tetrads of $\Sigma_{A,B}$ vanish.

In the **first stage** (Algorithm 2, see also Fig. 2), we identify clusters of observed variables with the same latent parent. However, note that since this stage only identifies *leaves* with the same latent parent, it is not guaranteed to identify all nodes with the same latent parent. This stage simultaneously recovers an ordering over these clusters. Thus, in the **second stage** (Algorithm 3, see also Fig. 3a), we iterate over pairs of clusters output from the first stage, identify pairs of clusters

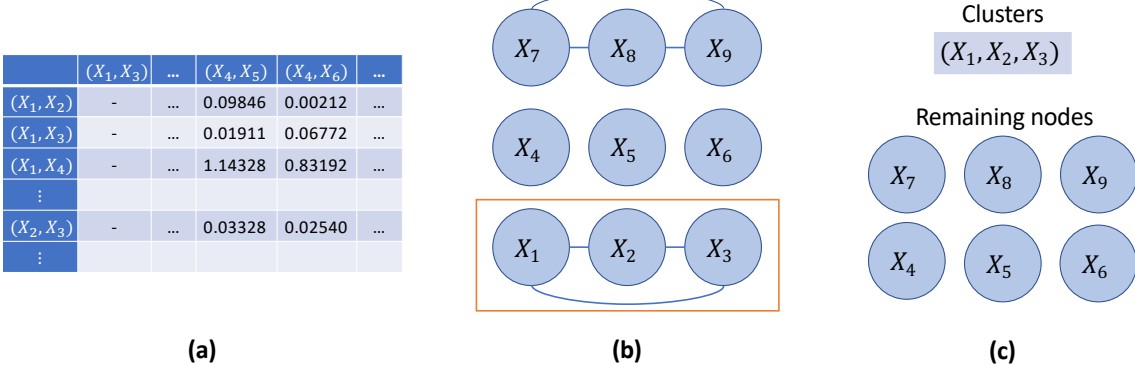

Figure 2: **Phase 1**: Let the true LFCM be the graph in Fig. 3(c). In our algorithm's first phase (see Algorithm 2) we perform the following steps: **(a)** Compute tetrad scores between all pairs of nodes. **(b)** For each pair of nodes, test the null hypothesis that all tetrads are zero. Construct a graph with the edge $i - j$ for any pair of nodes where we do not reject the null hypothesis. **(c)** Extract a clique from this graph to be a cluster (e.g., by picking the largest clique with arbitrary tie breaking), remove these nodes and repeat with remaining nodes.

with the same latent parent, and merge them, while leaving the ordering of the clusters intact. In the **third stage** (Algorithm 4, see also Fig. 3bc), we use the clustering and ordering information discovered in the previous two stages to learn a DAG over both latent and observed variables. In particular, given a node $X_j$ in cluster $C_j$ which comes before the cluster $C_i$ in our ordering, we wish to determine whether $X_j$ has an edge to the associated latent variable $L_i$. By Assumption 2, this can be accomplished by checking partial correlations between $X_j$ and the nodes in $C_i$. These stages compose a consistent algorithm, as established in the following theorem and proven in Appendix B. In Section 3, we have already discussed the importance of the first three structural assumptions from Definition 1. In Appendix C, we show how our algorithm fails under a violation of the double-parent assumption.

**Theorem 2** *Let $\mathcal{G}$ be a linear LFCM and let $\mathbb{X} \in \mathbb{R}^{n \times p}$ be a matrix of samples of the observed variables $X_1, \ldots, X_p$. Then Algorithm 1 is consistent under Assumptions 1, 2, and 3, i.e., as $n \to \infty$, we have $\mathbb{P}(\hat{\mathcal{G}} = \mathcal{G}) \to 1$.*

Next, we outline the complexity of our algorithm, using placeholders for the complexities of hypothesis tests in order to keep our results general. Let $f(d, n)$ denote the cost of performing the hypothesis test $H_{vt}$ on $d$ statistics from $n$ samples, when all sufficient statistics are pre-computed. Let $g(p)$ denote the complexity of an algorithm used to find a clique in a graph on $p$ nodes. While finding the *largest* clique in a graph is in general NP-hard, we can avoid this complexity, since the consistency of our algorithm does not rely on picking the largest clique at each step, only a clique of 3 or more nodes (picking larger cliques is simply a tool for improving statistical accuracy).

In the following, let $M$ be the maximum size of any returned cluster, and let $K$ be the number of clusters discovered by the algorithm. By definition, $M \le p$ and $K \le p$, so replacing these quantities by $p$ gives complexities that are only in terms of the known problem parameters. However, such upper bounds can be highly pessimistic. If the true graph has few nodes per cluster, or a small number of latent nodes, then with enough samples, $M$ and $K$ will also be small, respectively.

**Algorithm 2** `FindOrderedClusters`

**Input:** Data $\mathbb{X}$.
Initialize $R = [p]$
Initialize $\pi$ as an empty list
**while** $|R| > 3$ **do**
    Initialize $\mathcal{G}$ as an empty graph
    **for** *each pair of nodes $i, j$ in $R$* **do**
        if $H_{vt}(\{i,j\}, [p] \setminus \{i,j\})$, then add $i -$
        $j$ to $\mathcal{G}$
    **end**
    Let $C$ be the largest clique in $\mathcal{G}$, breaking
    ties arbitrarily
    $R = R \setminus C$
    Remove $C$ from the columns of $\mathbb{X}$
    Append $C$ to $\pi$
**end**
**if** $|R| > 0$ **then**
    Append $R$ to $\pi$
**end**
**Output:** An ordered clustering $\pi$

**Algorithm 3** `MergeClusters`

**Input:** Data $\mathbb{X}$, ordered clustering $\pi$
Repeat the following until convergence:
**for** *Clusters $c_1$ and $c_2$ such that $c_1 \prec c_2$* **do**
    **if** $H_{vt}(c_1 \cup c_2, c_1 \cup c_2)$ **then**
        Add nodes in $c_2$ to $c_1$ and delete $c_2$
    **end**
**end**
**Output:** An ordered clustering $\pi$

**Algorithm 4** `LearnDAG`

**Input:** Data $\mathbb{X}$, ordered clustering $\pi$
Initialize $\hat{\mathcal{G}}$ as an empty graph
Add $L_i$ to for each $C_i$ in $\pi$
Add $L_i \to X_j$ to $\hat{\mathcal{G}}$ for each $C_i$ in $\pi$, $X_j \in C_i$
**for** $X_j \prec L_i$ **do**
    Let $S = \cup_{k|L_k \prec L_i} \text{ch}(L_k)$
    Add $X_j \to L_i$ to $\hat{\mathcal{G}}$ if $H_{ci}(X_j, \text{ch}(L_i) \mid S)$
**end**
**Output:** DAG $\hat{\mathcal{G}}$

**Theorem 3** *The complexity of each algorithm is:*

(a) *Algorithm 2 takes* $\mathcal{O}(p^4 + p^3 f(p^2, n) + pg(p))$.

(b) *Algorithm 3 takes* $\mathcal{O}(p^2 M^4)$.

(c) *Algorithm 4 takes* $\mathcal{O}(pKM)$.

**Proof (a)** Computing all tetrads and their associated p-values is $\mathcal{O}(p^4)$. In each round, we perform $\mathcal{O}(p^2)$ hypothesis tests, each on $\mathcal{O}(p^2)$ statistics, so that the complexity at each round is $\mathcal{O}(p^2 f(p^2, n))$. After performing these tests, we identify the largest clique in a graph of $\mathcal{O}(p)$ nodes, so that the total run time per round is $\mathcal{O}(p^2 f(p^2, n) + g(p))$. At most $p$ rounds are required, resulting in the stated complexity.

**(b)** We must check $\mathcal{O}(p^2)$ pairs of clusters for whether or not they should be merged, and the maximum size of the union of any such pair is $\mathcal{O}(M)$. To check whether $\mathcal{O}(M)$ nodes belong to the same cluster, we require a hypothesis test on $\mathcal{O}(M^4)$ statistics, so that this step takes $\mathcal{O}(p^2 f(M^4, n))$.

**(c)** We perform $\mathcal{O}(pK)$ hypothesis tests, each based on $\mathcal{O}(M)$ partial correlations. ∎

Assume that we use the Sidak adjustment procedure explained in the next section (so that $f(d, n) = \mathcal{O}(d)$), and a greedy algorithm for picking cliques (so that $g(p) = p^3$). Then, replacing $K$ and $M$ by $p$, we have that Algorithm 2 takes $\mathcal{O}(p^5)$, Algorithm 3 takes $\mathcal{O}(p^6)$, and Algorithm 4 takes $\mathcal{O}(p^3)$, so that the overall complexity of our algorithm is at most $\mathcal{O}(p^6)$. Even in this pessimistic analysis, this complexity is relatively low for causal structure learning, which is known to be NP-hard in general (Chickering et al., 2004), and for which variants of the best-known algorithms, such

as PC (Spirtes et al., 2000) and GES (Chickering, 2002), typically have complexity $\mathcal{O}(p^{d+2})$, where $d$ is the maximum in-degree of the graph (Chickering, 2020).

## 4.1. Implementation Details

The null hypothesis $H_{vt}$ and $H_{ci}$ used in Algorithms 2, 3, and 4 imply that some vector-valued statistic of the covariance matrix is equal to zero. Procedures for *simultaneous hypothesis testing* are designed to (asymptotically) control the false discovery rate (FDR) of such a form of hypothesis test. In practice, we found that computing marginal p-values and performing *Sidak adjustment* Drton and Perlman (2007) yields good performance. In particular, given p-values $\{\pi_m\}_{m=1}^M$, the Sidak-adjusted p-values are

$$\pi_m^{\text{sidak}} = 1 - (1 - \pi_m)^M$$

**Remark 2** *The Sidak adjustment uses only the marginal distributions of each tetrad, neglecting potentially important information about the correlations between tetrads. In contrast, the* max-T *adjustment accounts for correlations between the tested statistics by estimating their correlation matrix, and has been shown to outperform the Sidak adjustment both theoretically and in practice (Drton and Perlman, 2007; Chernozhukov et al., 2013). However, the max-T adjustment requires sampling from a potentially high-dimensional multivariate normal distribution, an operation which is $\mathcal{O}(d^3)$ for dimension $d$. We have found that in practice, max-T adjustment performs similarly to Sidak adjustment while taking substantially longer. Therefore, we use Sidak adjustment for our experimental results, but provide capability for max-T adjustment in our codebase.*

Given a set adjusted p-values and a significance level $\alpha$, we reject the null hypothesis if *any* of the adjusted p-values are smaller than $\alpha$. To test conditional independence, recall that $H_{ci}(X_j, X_A \mid X_B)$ holds in a multivariate normal if and only if the vector of partial correlations $\{\rho_{ij|B}\}_{i \in A}$ is zero. To compute p-values, we use a widely used procedure which we call the *Fisher correlation test*. First, given the sample partial correlations $\{\hat{\rho}_{ij|B}\}_{i \in A}$, we apply the *Fisher z-transformation* $\hat{z}_{ij|B} = \sqrt{n - |B| - 3} \operatorname{arctanh}(\hat{\rho}_{ij|B})$. Then, we compute the two-tailed p-value of $\hat{z}_{ij|B}$ with respect to $\mathcal{N}(0, 1)$, i.e., $\pi_{ij|B} = 2Q(|\hat{z}_{ij|B}|)$, where $Q$ is the tail distribution function of $\mathcal{N}(0, 1)$.

Next, to test $H_{vt}(X_A, X_B)$, we adopt the widely-used *Wishart test* to compute the p-values (Wishart, 1928; Kummerfeld and Ramsey, 2016), which we now briefly describe. First, we compute the *sample tetrads* $\hat{t}_{ij,uv} = \hat{\Sigma}_{iu}\hat{\Sigma}_{jv} - \hat{\Sigma}_{iv}\hat{\Sigma}_{ju}$ for $\{i, j\} \subset A$ and $\{u, v\} \subset B$ such that $i, j, u$ and $v$ are distinct. Then, we normalize each sample tetrad, dividing by an estimate of its standard deviation to obtain the z-score $\hat{z}_{ij,uv}$. Drton et al. (2008) give the following formula for the variance of sample tetrads in terms of the true covariance matrix $\Sigma$:

$$\operatorname{Var}\left(\hat{t}_{ij,uv}\right) = n \cdot (n-1)^{-3} \cdot \left((n+2)|\Sigma_{[ij],[ij]}| \cdot |\Sigma_{[uv],[uv]}| - n|\Sigma_{[ijuv],[ijuv]}| + 3n|\Sigma_{[ij],[uv]}|\right),$$

where $|A| = \det(A)$. To estimate the variance, we use the above formula with the sample covariance $\hat{\Sigma}$ replacing $\Sigma$. Finally, we compute the two-tailed p-value of $\hat{z}_{ij,uv}$ with respect to $\mathcal{N}(0, 1)$.

## 5. Empirical Results

We evaluate our algorithm in two settings. First, we evaluate in a purely synthetic setting, which allows us to generate SCMs which exactly match our proposed model. Then, we evaluate in a semi-synthetic setting, modifying real data to more closely match our proposed model while demonstrating that our approach has promise in real-world biological settings.

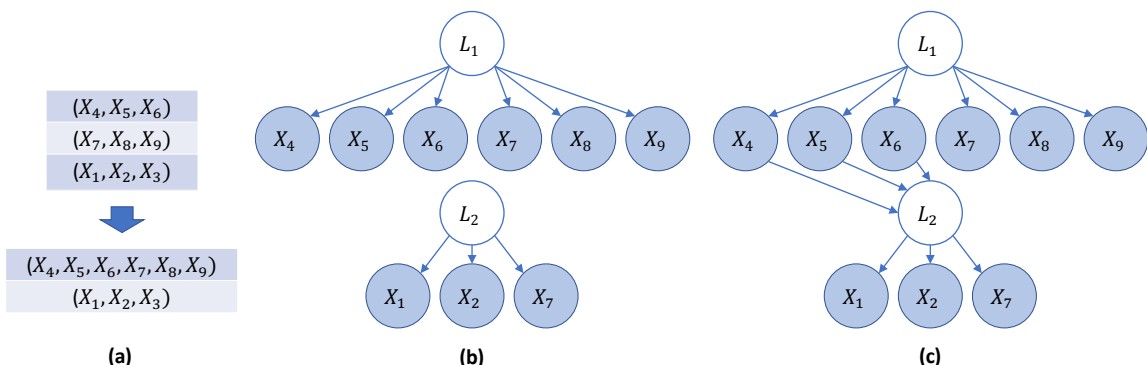

Figure 3: **Phases 2 and 3. (a)** Merge pairs of clusters based on vanishing tetrad tests. **(b)** Introduce latent nodes, and add edges from latent nodes to children. **(c)** Add parents of latent nodes based on conditional independence testing.

### 5.1. Synthetic data experiments

We begin by briefly describing the simulation settings used for our experiments, before describing the baselines and metrics which we use for evaluation. We generate a graph with 10 latent nodes, we first sample a "latent" skeleton $L(\mathcal{G})$ over $\{1, 2, \ldots, 10\}$ from a directed Erdös-Rényi model with edge probability 0.5. Then, for each latent node $L_k$, we generate $c_k \sim \mathsf{Unif}(3, 6)$ children. Finally, for each edge $L_k \to L_{k'}$ in $L(\mathcal{G})$, we sample $d_{k,k'} \sim \mathsf{Unif}(2, |\operatorname{ch}(L_k)|)$, then sample $d_{k,k'}$ children of $L_k$. For each selected child $c_k$, we add the edge $c_k \to L_{k'}$, giving us a DAG $\mathcal{G}$ over both latent and observed nodes which has latent skeleton $L(\mathcal{G})$ and satisfies Definition 1.

Given this DAG, we generate a linear SCM as follows, proceeding in topological order. For each node $X_j$, if the node has no parents, its equation is $X_j = \epsilon_j$ for $\epsilon_j \sim \mathcal{N}(0, 1)$. If the node has parents, then for each parent $X_i$, we sample an "initial" weight $\tilde{w}_{ij} \sim \mathsf{Unif}([-1, -.25] \cup [.25, 1])$. Next, we describe how to normalize these weights in order to avoid the *varsortability* issue described by Reisach et al. (2021), where simulated DAGs are easy to learn because the variance of each node tends to increase according to the topological order. Given these initial weights, we simulate $B$ "parental contributions" $\mu_j^{(b)} = \sum_{i \in \mathrm{pa}_\mathcal{G}(j)} \tilde{w}_{ij} X_i^{(b)}$ for $X^{(b)}$ sampled from the linear SEM defined over $i < j$. The sample variance $\hat{\sigma}_j$ of $\mu_j^{(b)}$ serves as an estimate for the variance that the parents of $j$ will contribute to $X_j$. Finally, we ensure that $X_j$ has variance 1 and that half of its variance is contributed by its parents by setting the final weights as $w_{ij} = (2\hat{\sigma}_j)^{-1/2} \tilde{w}_{ij}$ and $\epsilon_j \sim \mathcal{N}(0, 1/2)$.

**Accuracy of learning clusters.** In our first set of experiments, we evaluate the accuracy of the learned clusters. To measure the accuracy over the learned clustering compared to the underlying clustering, we use the following criteria: the pair $(X_i, X_j)$ is a *true positive* if $X_i$ and $X_j$ are in the same underlying cluster and are in the same learned cluster, the pair is a *false positive* if $X_i$ and $X_j$ are not in the same underlying cluster but are in the same learned cluster, and so on. We generate 50 different SEMs via the process described above, and from each SEM we generate $n = 200$ samples. We run our algorithm using significance levels ranging from .05 and .5. The results are shown in Fig. 4a. Due to interactions between the hypothesis tests used by our algorithm (denoted "LFCM", shown in blue), the ROC curve is highly non-monotonic over larger ranges of values, so that the curve occupies only a small range of the plot, though it clearly drastically outperforms the competing methods, achieving almost perfect performance. In particular, we consider two baselines. First,

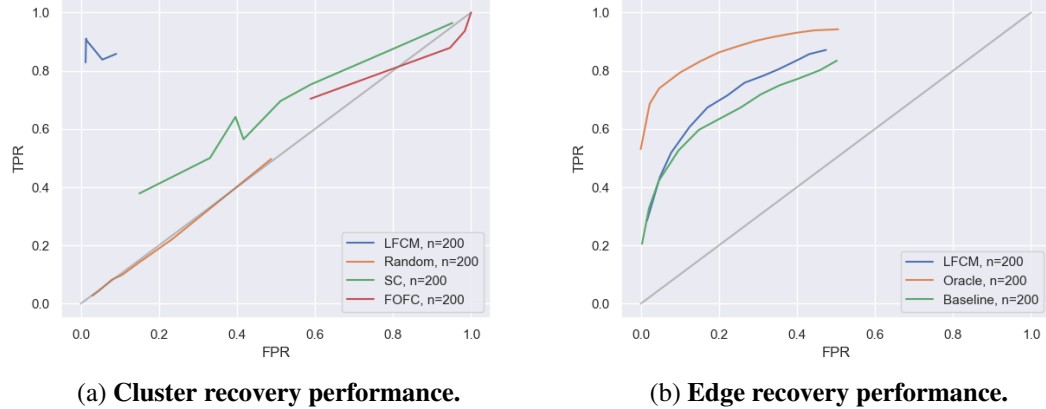

(a) **Cluster recovery performance.**  (b) **Edge recovery performance.**

Figure 4: **Performance on synthetic data.** The first two phases of our algorithm almost perfectly recover the ground truth clusters, while the third phase of our algorithm demonstrates the utility of multiple hypothesis testing for recovering edges between observed nodes and latent nodes.

we compare to *spectral clustering* (denoted "SC", shown in green), as implemented in `sklearn` in Python, a widely used clustering technique in genomics (Higham et al., 2007), with a varying number of estimated clusters from 2 to 30. Second, we compare to the FindOneFactorCluster algorithm of Kummerfeld and Ramsey (2016) (denoted "FOFC", shown in red). We found that spectral clustering performs slightly better than random guessing, while FOFC performs about the same as random guessing, reflecting the drastic deviation from the measurement assumption on which it relies. Finally, we verified that randomly picking $K$ clusters of equal size (denoted "Random", shown in orange), for $K$ varying from 2 to 30, matched the diagonal random guessing line.

**Accuracy of learning edges from observed nodes to latent nodes.** In our second set of experiments, we evaluate the accuracy of learning the edges from observed nodes to latent nodes, when the true clusters and their ordering is known. In particular, for $L_i \prec L_j$ in the ordering, and $X_i \in \text{ch}(L_i)$, the pair $(X_i, L_j)$ is considered a *true positive* if $X_i \to L_j$ in the true LFCM as well as in the estimated LFCM, a *false positive* if it is not in the true LFCM but does appear in the estimated LFCM, and so on. In Fig. 4b, we compare the third phase of our algorithm (denoted "LFCM", shown in blue) to a *baseline* which simply uses a single child of each latent node for the conditional independence test (denoted "Baseline", shown in green), as well as an *oracle* which is able to observe the values of the latent nodes and is thus infeasible (denoted "Oracle", shown in orange). As expected, our algorithm does not perform as well as this unrealizable case, but still performs significantly better than random (the diagonal line) and noticeably better than the baseline.

## 5.2. Semi-synthetic experiments on protein signaling data

In this section, we demonstrate the applicability of our method to a real-world dataset in a semi-synthetic setting. The Sachs protein mass spectroscopy dataset (Sachs et al., 2005) is a widely used benchmark for causal discovery, in part due to the existence of a commonly accepted ground truth network over the 11 measured protein expression values, shown in Fig. 5a. We use the 1,755 "observational" samples, where the experimental conditions involve only perturbing receptor enzymes,

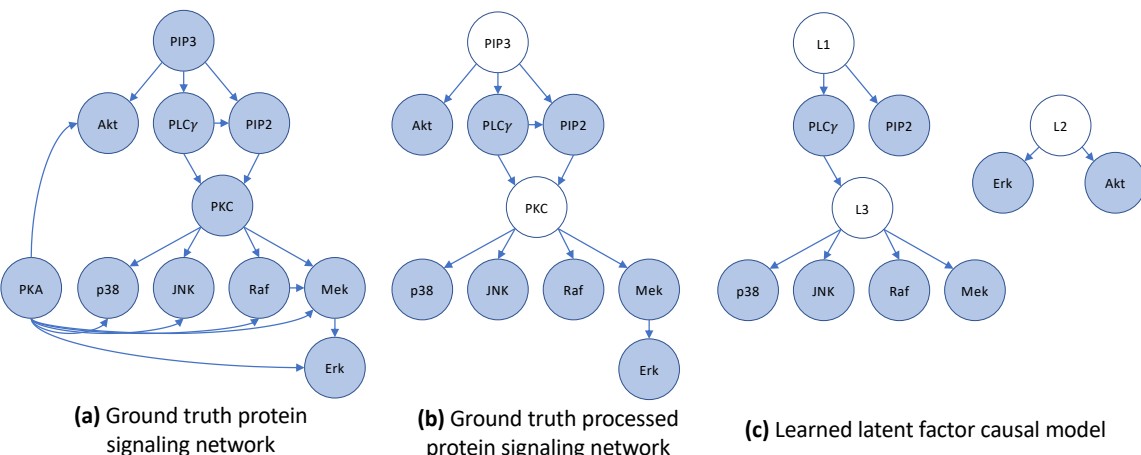

**(a)** Ground truth protein signaling network

**(b)** Ground truth processed protein signaling network

**(c)** Learned latent factor causal model

Figure 5: **Learning a latent factor causal model for protein signaling**. Our recovered model in **(c)** nearly captures the ground truth network in **(b)**.

and not any signaling molecules, as described in Wang et al. (2017). To make the ground truth network more similar to a latent factor causal model, we perform three data-processing steps: (1) we "condition" on PKA, by regressing it out of the dataset, (2) we "remove" the direct effect of Raf on Mek, and (3) we "marginalize" out PIP3 and PKC by removing the corresponding columns from the dataset. We "remove" the direct effect of Raf on Mek as follows. First, we regress Mek on its two remaining parents, Raf and PKC. Call the resulting regression coefficient for Raf $\beta_{Raf}$. For each sample, we subtract the value of Raf times the $\beta_{Raf}$ from the value of Mek. Note that we do *not* remove the direct effect of PLC$\gamma$ on PIP2, since then our algorithm collapses all nodes into a single cluster. The processed graph is show in Fig. 5b.

Running our method with significance level $\alpha = 0.01$ for $H_{vt}$ and $\alpha = 0.1$ for $H_{ci}$, we obtain the network shown in Fig. 5c. The clustering by our algorithm closely matches the clustering (*Akt, PLC$\gamma$, PIP2*), (*p38, JNK, Raf, Mek, Erk*) induced by the true network, with the exception that *Akt* from the first cluster and *Erk* from the second cluster are pulled out into a cluster with one another, which may indicate that the effect of PKA on Akt and Erk cannot be completely removed using a purely linear approach. The ordering between the clusters (*PLC$\gamma$, PIP2*) and (*p38, JNK, Raf, Mek*) is preserved, but the edge $PIP2 \rightarrow L3$ is missing.

## 6. Discussion

In this paper, we introduce a method (Algorithm 1) for learning *latent factor causal models* (LFCMs), a novel, biologically-motivated class of causal models with latent variables. We showed that these models are identifiable in the linear setting using rank constraints on submatrices of the covariance matrix, and that our method provides a consistent estimator for these models. We also showed that our method outperforms existing clustering algorithms on synthetic data, and almost perfectly re-covers a widely-accepted ground truth network in a semi-synthetic biological setting. These results serve as a proof-of-concept, suggesting that our algorithm may be able to shed biological insight on the problem of identifying the spatial clustering of genes in the cell nucleus given data on the expression of the genes. Interestingly, since it is possible (although expensive) to measure the 3D

organization of the genome in the cell nucleus (Lieberman-Aiden et al., 2009), there is a meaningful avenue to validate our method on the important biological application of connecting 3D genome organization with gene expression (Uhler and Shivashankar, 2017). We conclude with a discussion of the limitations of our model, which suggests a number of other directions for future work.

**Limitations.** The latent factor causal model (LFCM) class considered in this paper has two obvious limitations. First, we make the strong parametric assumption of a linear Gaussian SEM. While many nonparametric conditional independence tests have been proposed (Gretton et al., 2007; Zhang et al., 2011), we are not aware of nonparametric tests for shared latent factors that would generalize $H_{vt}$. Thus, extending our algorithm to a nonparametric setting would require development of such tests. In particular, generalizing tetrad constraints to the nonlinear setting is an interesting direction for future research. Second, we make two strong structural assumptions. The "unique cluster" assumption is well-motivated by our biological setting of interest, and is likely the easiest assumption to remove since Theorem 1 already provides a generalization of the rank constraint we leverage. Indeed, generalized rank constraints have already been explored in prior work on factor analysis (Drton et al., 2007; Kummerfeld et al., 2014). The "bipartite assumption" has two components which may be separately examined. First, the assumption that there are no edges between observed variables is most reasonable for systems such as gene regulatory networks where a different, unobserved entity class (in this case, proteins) mediates *all* interactions between the observed variables (i.e., genes). This assumption may also be expendable, for instance by allowing for a *small number* of edges between observed variables, akin to the low-rank plus sparse literature in previous work on learning with exogenous latent variables (Frot et al., 2019; Agrawal et al., 2021). Similarly, existing techniques (Cai et al., 2019; Xie et al., 2020) may help to eliminate the assumption that there are no edges between latent variables. In addition to relaxing these assumptions, it would be of interest to develop procedures for testing these assumptions in data, e.g., by extending recent work (Agarwal et al., 2020) which develops a spectral-energy-based hypothesis test for structural assumptions in latent variable models.

## Acknowledgments

We would like to thank Louis Cammarata for many helpful discussions. Chandler Squires was partially supported by an NSF Graduate Research Fellowship. All authors were partially supported by NSF (DMS-1651995), ONR (N00014-17-1-2147 and N00014-22-1-2116), the MIT-IBM Watson AI Lab, MIT J-Clinic for Machine Learning and Health, the Eric and Wendy Schmidt Center at the Broad Institute, and a Simons Investigator Award to Caroline Uhler.

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

## Appendix A.  Faithfulness assumptions are generic

We first recall the assumptions from Section 3.

**Assumption 1 (Cluster tetrad faithfulness)** *Suppose $X_i$ and $X_j$ are not in the same cluster. Then there exists some $\{u, v\}$ such that $t_{ij,uv} \neq 0$.*

**Assumption 2 (Parent tetrad faithfulness)** *Suppose $X_i$ and $X_j$ are in the same cluster, but $X_i$ has at least one child. Then there exists some $\{u, v\}$ such that $t_{ij,uv} \neq 0$.*

**Assumption 3 (Latent adjacency faithfulness)** *Suppose $X_i \rightarrow L_k$. Let $S_i = \text{ch}(\text{pa}(X_i)) \setminus \{i\}$ and $S' = \cup_{j \leq i} \text{ch}(L_i)$. Then $\rho_{i,k|S_i,S} \neq 0$ for some $X_k \in \text{ch}(L_k)$*

**Proposition 1** *Assumption 1 holds generically under the first three assumptions in Definition 1.*

**Proof** Let $L_i = \text{pa}(X_i)$ and $L_j = \text{pa}(X_j)$. By the triple child assumption, there exists some $X_u$ in the same cluster as $X_i$, and some $X_v$ in the same cluster as $X_j$. Then any set which t-separates $\{i, j\}$ and $\{u, v\}$ must contain $L_i$ and $L_j$, since $i$ must be separated from $u$ and $j$ must be separated from $v$, respectively. Therefore, by Theorem 1, $\text{rank}(\Sigma_{[ij],[uv]}) = 2$ generically, i.e., $t_{ij,uv} \neq 0$ generically. ∎

**Proposition 2** *Assumption 2 holds generically under the first three assumptions in Definition 1.*

**Proof** Let $L_i = \text{pa}(X_i) = \text{pa}(X_j)$. $L_i$ must have some other child $X_u$ by the triple child assumption. Let $L_v$ be some child of $X_i$, and $X_v$ be some child of $L_v$. Then any set which t-separates $\{i, j\}$ and $\{u, v\}$ must contain $L_i$ and $L_v$, since $i$ must be separated from $j$ and $i$ must be separated from $v$, respectively. Therefore, by Theorem 1, $\text{rank}(\Sigma_{[ij],[uv]}) = 2$ generically, i.e., $t_{ij,uv} \neq 0$ generically. ∎

**Proposition 3** *Assumption 3 holds generically under the first three assumptions in Definition 1.*

**Proof** If $X_i \rightarrow L_k$, then $X_i$ and $X_k$ are d-connected given $S_i, S$ for any $X_k \in \text{ch}(L_k)$. Spirtes et al. (2000) establish that if two nodes are d-connected, then their partial correlation in a linear SEM is generically nonzero, proving the desired result. ∎

## Appendix B.  Proof of Theorem 2

**Theorem 2** *Let $\mathcal{G}$ be a linear LFCM and let $\mathbb{X} \in \mathbb{R}^{n \times p}$ be a matrix of samples of the observed variables $X_1, \ldots, X_p$. Then Algorithm 1 is consistent under Assumptions 1, 2, and 3, i.e., as $n \rightarrow \infty$, we have $\mathbb{P}(\hat{\mathcal{G}} = \mathcal{G}) \rightarrow 1$.*

**Proof** By Assumption 1 and Assumption 2, as long as at least two nodes are present from each cluster, if $t_{ij,uv} = 0$ for all pairs $u, v \in \{i, j\}$ iff. $X_i$ and $X_j$ have the same latent parent, and neither node has any children. Thus, if $i - j$ in $\mathcal{G}$ in Algorithm 2, then $i$ and $j$ are in the same cluster. Next, if $i$ and $j$ are both left in $R$ after the while loop, then they must be in the same cluster.

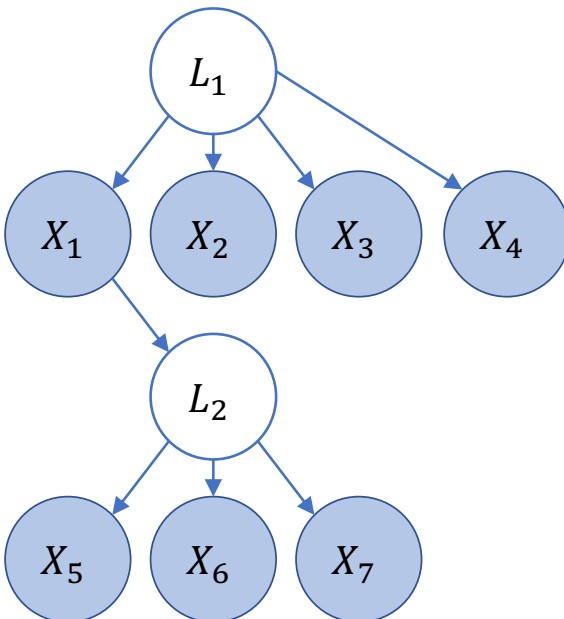

Figure 6: Algorithm 1 can fail when the true graph violates the double parent assumption.

For sake of contradiction, suppose not, and let $L_i = \mathrm{pa}(X_i)$, $L_j = \mathrm{pa}(X_j)$. Since $X_i$ remains, then by the double-parent assumption, there must also remain some other node $X_{i'}$ that is a child of $L_i$. Similarly, there must remain some other node $X_{j'}$ that is a child of $L_j$. However, then $|R| = 4$, a contradiction. Therefore, the clustering output by Algorithm 2 is a refinement of the true clustering. Furthermore, if a node $i$ is upstream of the cluster $C_1$, then $i$ necessarily has a child, and thus $t_{ij,uv} \neq 0$ for some $j, u, v$. Therefore, $i$ cannot be placed in any clique before the cluster $C_1$ is completely removed, and thus the ordering of cluster returned by Algorithm 2 is topologically consistent. By the double parent assumption, each cluster in $\pi$ from Algorithm 2 has size at least 2. Assumption 1 ensures that two clusters are merged by Algorithm 3 iff. they have the same latent parent. Finally, Assumption 2 ensures that $X_i \to L_k$ in $\hat{\mathcal{G}}$ iff. $X_i \to L_k$ in $\mathcal{G}$. ∎

## Appendix C. Double-parent violation

Let $\mathcal{G}$ be the graph in Fig. 6. Suppose Algorithm 2 first removes the cluster $\{X_2, X_3, X_4\}$. Then, $X_1$ is only t-separated by from $\{X_5, X_6, X_7\}$ by $L_2$, so in the second round of Algorithm 2, it is clustered with these nodes instead of with $\{X_2, X_3, X_4\}$.

