# OpenReview forum: "Causal Structure Discovery between Clusters of Nodes Induced by Latent Factors"
_cclear.cc/CLeaR/2022/Conference — CLeaR 2022 Poster_

### Official Review · Reviewer_nsAC · 2021-11-18

**Confidence:** 4
**Overall Score:** 6

**Main Review:**

The authors proposed a latent factor causal model with some restricted assumptions, which was motivated by gene regulatory networks. And they proposed a consistent three-stage algorithm to identify such a model. Overall, I think the paper is written with some minor mistakes, but the motivation from the biology field is interesting. I have some concerns listed in the following.

1. The identifiability seems not to be clear. What actual identifiability results did they achieve? And to what extent they identify? Did the overall structure including the edges from the observed to the latent could be identified, or only the cluster? It is better to make them clarified.

2. As for the [Double-parent assumption] in Definition 1, what would happen if there is only one observed variable X_i in C_k that points to L_{k’}? May be better to explain it and add this case in the synthetic experiments.




**Summary:**

The authors proposed a latent factor causal model with some restricted assumptions, which was motivated by gene regulatory networks. And they proposed a consistent three-stage algorithm to identify such a model.

---

> ### Author Response · Authors · 2021-12-03
> **Response to nsAC**
>
> Thank you for the useful review!
>
> ## Identifiability
> Please see our response to Reviewer xSua. Briefly, the fact that our algorithm is consistent constitutes a constructive proof that the graph, including the latent variables, can be exactly identified under the model assumptions.
>
> ## Double-parent assumption
> Thank you for pointing out an opportunity to better clarify the necessity of our assumptions. We will add an example to the paper of how the algorithm fails if the double-parent assumption is violated. In particular, say that the double parent assumption is violated, with $X_i$ in one cluster being the only node pointing to $L_j$, the latent parent of another cluster. Let $X_j, X_u, X_v$ be children of $L_j$. Then $(X_i, X_j)$ may be t-separated from $X_u, X_v$ by $L_j$ alone, so that $t_{ij,uv} = 0$, in which case $X_i$ may be incorrectly put into the cluster belonging to $L_j$.

---

> > ### Comment · Reviewer_nsAC · 2021-12-12
> > **Further comment**
> >
> > Thanks for the authors' response.
> >
> > As for the Double-parent assumption, it might be better to provide some synthetic experiments where the assumptions are violated or at least add more discussions in Section 6.
> >
> > Overall, though I think this model has some strong assumptions, its motivation by gene regulatory networks based on the latent factor model is still interesting and their model may attain the potential application ability in discovering causal relations between clustered genes in the biological domain. Thus, it is acceptable for me to increase my score to 6.

---

### Official Review · Reviewer_bvqT · 2021-11-21

**Confidence:** 3
**Overall Score:** 6

**Main Review:**

Pros:
(1)	The addressed problem is realistic and significant.
(2)	Experiments results are convincing.
Cons/Questions:
(1)	The motivation is still not very clear. Specifically, the authors state that they would address the problem of causal discovery in the presence of unobserved confounders. They carefully discuss the two types of methods for handling this problem. However, why is the proposed method better? Could it address some issues of existing methods? In the current version, the answers are not clear. Therefore, it is a bit hard to understand the motivation and contribution of this paper well. Such explanations are important and should be added.
(2)	This paper argues that one of the weaknesses of the existing methods is the rationality of assumptions. Nevertheless, it seems that the method in this paper needs more assumptions. How to check them? For Definition 1 and Assumption 1, 2, 3, there are not enough analyses, e.g., the application scenario.
(3)	Could the authors add more details for Definition 1? For example, the discussions on Definition 1(b), the relationship between edges and the rank of sub-matrix.
(4)	Is the complexity of the method in this paper is relatively high?
(5)	The descriptions of experiments can be improved.
-	The descriptions of experimental settings are less logical, especially in Section 5.1.
-	Discussions on experimental results are insufficient.

In summary, this paper is appreciated. However, there are major issues/concerns listed above.


**Summary:**

This paper tackles the problem of causal discovery in the presence of latent variables. Motivated by gene regulatory networks, the authors design a three-stage algorithm that is consistent under some conditions. Theoretical analyses are provided for the identiﬁability of results. Empirical evaluations are provided to verify the effectiveness of the proposed method in this paper.

---

> ### Author Response · Authors · 2021-12-03
> **Response to bvqT**
>
> Thank you for the helpful review!
>
> ## Motivation and Assumptions
> First, we want to clarify that our method is only intended to be better **in certain settings**, such as the genomic setting that we use as motivation. In such settings, observed variables can be parents of latent variables. As we point out at the end of the section “Learning causal models with latent variables”, this breaks the measurement assumption used by a majority of existing methods. We will include an example of what these methods would do in our setting: each true latent variable would be “exogenized” (see “Graphs for margins of Bayesian networks” (Evans 2014)), a process which adds a potentially large number of edges between observed variables.
>
> We also discuss methods such as FCI which learn an equivalence class of “ancestral graphs”. However, the ancestral graphs for the setting we consider would also lead to dense networks over the observed variables, and the edges between observed variables would not be directed. We will also include an example of what these algorithms do in our setting for further clarity.
>
> Accordingly, we did not intend to argue that the weakness of existing methods was the rationality of the assumptions in all settings. Rather, the different methods, including ours, are tailored to different settings, and to capture different information. Because we do not make the measurement assumption, and we are not satisfied with the information captured by ancestral graphs, we must introduce new assumptions, such as the ones in Definition 1 (Assumptions 1-3 are less significant - faithfulness assumptions such as these are standard necessities for “dotting your i’s and crossing your t’s” in proofs). As discussed in the Limitations section, we are aware of the drawbacks imposed by these assumptions, and suggest avenues for removing these assumptions. We will also add references to possible ways of testing these assumptions, such as by extending recent work (e.g., “Synthetic Interventions” by Agarwal et. al.) which tests for structural assumptions on latent variable models using spectral information.
>
> ## Details in Definition 1
> Yes, we will add a description after Theorem 1, after the necessary concepts are introduced, to clarify how the bipartite assumption in Definition 2 relates to sub-matrix ranks. In particular, if two observed variables $i$ and $j$ have an edge between them, then Theorem 1 says that the rank of any submatrix which includes those two variables in both the rows and the columns will generically be of rank 2, since any t-separating sets $C_A, C_B$ must include $i$ and $j$. This would prevent the variables from being merged into a single cluster if they are in separate clusters after the first phase of the algorithm.
>
> We will note that this assumption could therefore be removed and replaced by an assumption that all observed variables in a cluster have directed edges into the next cluster, a stronger form of the double-parent assumption.
>
> ## Complexity
> The complexity of our method is actually relatively low for causal structure discovery. If we pessimistically take $K = M = p$, (where $K$ is the number of clusters, $M$ is the maximum cluster size, and $p$ is the number of variables), and we use Sidak adjustment (in which case $f(d, n) = \mathcal{O}(d)$), then our method is $\mathcal{O}(p^5)$. This is equal to the complexity of the PC algorithm when only conditioning sets up to size 3 are considered, which is only consistent when 1) there are no latent variables and 2) the maximum in-degree of the true DAG is less than or equal to 3, i.e., when the network is very sparse. We will add these explanations to our paper to describe how the computational complexity of our algorithm compares to that of other causal structure learning methods.
>
> ## Descriptions of experiments
>
> We would be very happy to clarify the descriptions of our experiments. Are there any specific parts of the description that you found unclear? This would help us improve the section.

---

### Official Review · Reviewer_xSua · 2021-11-22

**Confidence:** 4
**Overall Score:** 6

**Main Review:**

The paper tackles a non-trivial task and proposes a novel method to estimate the causal structure under their problem setting.
The paper builds on prior work in the field of Tetrad-based latent variables models. The authors put some effort into dropping the traditional measurement assumption and estimating the causal structure, including causal direction.
The authors have an in-depth understanding of the related works and provide a detailed review.


However, there are some concerns and suggestions,

1. The authors say that the casual order between clusters can be identified. Are there any theories to support this result? As far as I know, in a linear Gaussian setting, there is no possibility to detect the causal direction between variables, except the V-structure and Meek rules.


2. The example of In Section 4: The Phase I of the algorithm firstly selects the cluster \{ X_1, X_2, X_3 \}. What is the reason? Why not is the \{ X_4, X_5, X_6 \}? It seems that those two clusters have the same Tetrad constraints.

Besides, the authors should firstly give the ground-truth graph.
It would be helpful to describe the details of the example in the main graph.

3. In Section 5.2: How do you remove the direct effect of Raf on Mek? Why don't you do the same process for the edge between PLCy and PIP2?


minor typos:
Figure 3: there are two variables X_7 subgraph (b) and (c);

If the authors sufficiently address the mentioned concerns I am happy to change my assessment.

**Summary:**

Estimating causal structure area challenge in causality, especially in presence of unobserved variables. The authors first define a particular latent causal model, latent factor causal models, allowing the edges between observed variables and latent variables. Then, the authors propose a three-step method to estimate the model by making use of Tetrad constraints.  Although the work relies on strong assumptions, it appears to be both novel and significant.

---

> ### Author Response · Authors · 2021-12-03
> **Response to xSua**
>
> Thank you for the helpful review!
>
> ## Identifying Causal Order
> Thanks for your comment - we realize now that our identifiability result was not clearly emphasized. The short answer is **yes** - our identifiability result is theoretically supported, our proof is in Appendix B. In the final version, we will clarify that when we say that our Algorithm is consistent in Theorem 2, we mean that it **completely** identifies the graph, i.e., the order between the clusters, as well as the edges between latent and observed variables.
>
> As you noted, this differs from the setting where (1) the structural equations are linear Gaussian and (2) all variables are observed and no special structure is imposed. Our setting differs on point (2). In particular, the assumptions that each observed variable has a single latent parent, and that each latent variable has at least three children, allow us to guarantee being able to identify the bottom-most cluster(s). We then marginalize over the bottom-most cluster and repeat this process to build an order over clusters (Algorithm 2).
>
> ## Example in Figure 2
> We assume that the two clusters you refer to with the same tetrad constraints are $\\{ X_1, X_2, X_3 \\}$ and $\\{ X_7, X_8, X_9 \\}$ - as seen in Fig. 2b, the cluster $\\{ X_4, X_5, X_6 \\}$ does **not** have the same tetrad constraints.
>
> If this interpretation is correct, then you are right, the algorithm can pick either cluster (e.g., selecting between the two at random), or we could have opted to select all clusters at each step. If we instead picked $\\{ X_7, X_8, X_9 \\}$, the initial ordering between clusters would be different, but the ordered clustering after Phase 2 - the “merging” step in Algorithm 3 - would be the same, leading to the same final result.
>
> We will clarify that this “tie-breaking” does not affect the final result. The true graph used in our examples is given in Fig 3c as the final output of the algorithm; we will move this to come before Fig 2.
>
> ## Removing the effect of Raf on Mek
> Thanks for pointing out this detail which we forgot to include. To remove the effect of Raf on Mek, we perform linear regression of Mek on its parents, then we subtract the value of Raf times its corresponding regression coefficient from Mek. If the data were indeed generated by a linear structural causal model, and we were given the true regression coefficients, this would exactly correspond to “removing the direct effect” of Raf on Mek, i.e., setting the corresponding edge parameter to 0.
>
> When we remove the effect of PLCg and on PIP2, all nodes collapse into a single cluster. We intended to note this in the submission and will add a line describing this result.
>
> ## Minor typos
> Thanks for pointing these out!

---

> > ### Comment · Reviewer_xSua · 2021-12-11
> > **Additional question**
> >
> > Thanks for your clarification! Regarding the identification of causal order:  Consider the two structures (1) $\{X_1,X_2,X_3\} \to L_1 \to \{X_4,X_5,X_6\} $ and (2) $ \{X_4,X_5,X_6\}  \to L_1 \to \{X_1,X_2,X_3\}$. Does these two structures satisfy your setting? If they satisfy, how to distinguish them?

---

> > > ### Author Response · Authors · 2021-12-11
> > > **Response to additional question**
> > >
> > > Hi, thanks for taking the time to go over the response. This is a great question for elucidating what's going on.
> > >
> > > First, our structures always require each observed variable to have a latent parent. So, I would replace your structures with (1') $L_1 \to X_1, X_2, X_3 \to L_2 \to X_4, X_5, X_6$ and (2') $L_1 \to X_4, X_5, X_6 \to L_2 \to X_1, X_2, X_3$
> > >
> > > These two structures are distinguished by the ranks of certain submatrices: in structure (1') $\Sigma_{12,3456}$, $\Sigma_{13,2456}$, $\Sigma_{23,1456}$ are (generically) rank 2, while in structure (2'), all of these submatrices are rank 1. This is because in structure(1), the sets $A = \\{ X_1, X_2 \\}$ and $B = \\{ X_3, X_4 \\}$ are only t-separated by the set $\\{ L_1, L_2 \\}$ in structure (1'), but they are t-separated by the set $\\{ L_2 \\}$ in structure (2'). The result then follows from Theorem 1.

---

> > > > ### Comment · Reviewer_xSua · 2021-12-12
> > > > **Feedback**
> > > >
> > > > Thanks for your additional clarification- this makes a lot more sense.
> > > >
> > > > Overall, though their setting is restricted, this is a good start to recover causal structures without the traditional measurement assumption (allowing observed variables to affect latent variables). I will keep my overall score.

---

### Decision · Program_Chairs · 2022-01-12

**Decision:**

Accept (Poster)

**Comment:**

The authors propose a method for recovering a DAG from observational data with latent variables. For this to be possible, the authors must impose very strong assumptions on the structure of the latent and the observed variables. For instance, the authors assume no direct effects between the observed variables and no direct effects between the latents. They also make strong parametric assumptions (linear Gaussian SEM). The plausibility of these assumptions in general settings is questionable. However, the authors claim that their work is motivated by gene regulatory networks. The paper is well-written, the authors provide thorough references, and both theoretical and applied results are provided.

In my opinion, the work presented in this paper is important even though it focuses on a very narrow set of cases. Research into the causal discovery under some assumptions on the structure of latents is necessary to incorporate various kinds of domain-specific expert knowledge. As such, I am inclined to suggest this work for publication and see it as a starting point for future research on similar topics.